# Extracellular-Signal Regulated Kinase: A Central Molecule Driving Epithelial–Mesenchymal Transition in Cancer

**DOI:** 10.3390/ijms20122885

**Published:** 2019-06-13

**Authors:** Monserrat Olea-Flores, Miriam Daniela Zuñiga-Eulogio, Miguel Angel Mendoza-Catalán, Hugo Alberto Rodríguez-Ruiz, Eduardo Castañeda-Saucedo, Carlos Ortuño-Pineda, Teresita Padilla-Benavides, Napoleón Navarro-Tito

**Affiliations:** 1Laboratorio de Biología Celular del Cáncer, Facultad de Ciencias Químico Biológicas, Universidad Autónoma de Guerrero, Av. Lázaro Cárdenas s/n Chilpancingo, Gro. 39090, Mexico; monse-olea@live.com.mx (M.O.-F.); miriamzuniga@uagro.mx (M.D.Z.-E.); ecastaneda@uagro.mx (E.C.-S.); 2Laboratorio de Biomedicina Molecular, Facultad de Ciencias Químico Biológicas, Universidad Autónoma de Guerrero, Av. Lázaro Cárdenas s/n Chilpancingo, Gro. 39090, Mexico; mamendoza@uagro.mx (M.A.M.-C.); hugordzrz@outlook.com (H.A.R.-R.); carlos2pineda@hotmail.com (C.O.-P.); 3Department of Biochemistry and Molecular Pharmacology, University of Massachusetts Medical School, 364 Plantation Street, Worcester, MA 01605, USA

**Keywords:** extracellular-signal regulated kinase (ERK), epithelial–mesenchymal transition (EMT), cancer, metastasis, phosphorylation

## Abstract

Epithelial–mesenchymal transition (EMT) is a reversible cellular process, characterized by changes in gene expression and activation of proteins, favoring the trans-differentiation of the epithelial phenotype to a mesenchymal phenotype. This process increases cell migration and invasion of tumor cells, progression of the cell cycle, and resistance to apoptosis and chemotherapy, all of which support tumor progression. One of the signaling pathways involved in tumor progression is the MAPK pathway. Within this family, the ERK subfamily of proteins is known for its contributions to EMT. The ERK subfamily is divided into typical (ERK 1/2/5), and atypical (ERK 3/4/7/8) members. These kinases are overexpressed and hyperactive in various types of cancer. They regulate diverse cellular processes such as proliferation, migration, metastasis, resistance to chemotherapy, and EMT. In this context, in vitro and in vivo assays, as well as studies in human patients, have shown that ERK favors the expression, function, and subcellular relocalization of various proteins that regulate EMT, thus promoting tumor progression. In this review, we discuss the mechanistic roles of the ERK subfamily members in EMT and tumor progression in diverse biological systems.

## 1. Introduction

Epithelial–mesenchymal transition (EMT) is a process of trans-differentiation by which epithelial cells change to a mesenchymal phenotype [1]. At the biochemical level, a metabolic reprogramming of lipids and carbohydrates is closely related to the activation of the EMT program [2]. In the molecular context, the cells undergo changes in gene expression, function, and/or activation of proteins required for this transition [3]. These changes favor an increase in cell migration, invasion, and resistance to anoikis and chemotherapy [3]. The cytoskeleton is also reorganized, allowing three-dimensional movement of epithelial cells into the extracellular matrix (ECM) [3]. Thus, EMT gives cells the ability to migrate and invade adjacent or distal tissues, a classic feature of the phenotype of invasion or metastasis of tumor cells [4,5]. In recent years, the number of cancer cases has increased; cancer is well-known to be a leading cause of death worldwide [6]. 

The extracellular-signal regulated kinase (ERK) participates in many physiological and pathological processes such as proliferation, cell migration, invasion, and metastasis [4,5]. Two decades ago, ERK was implicated for the first time in EMT; since then, extensive research described the participation of ERK in this process using different models of study [7]. In this review, we focus on recent scientific advances that establish the role of ERK in both EMT and tumor progression in human cancer.

## 2. Epithelial–Mesenchymal Transition

EMT is a process of trans-differentiation by which epithelial cells change to a mesenchymal phenotype [1]. In higher eukaryotes, EMT is necessary for normal development, occurring by three different mechanisms with different functional results. Type 1 EMT takes place during implantation, embryogenesis, and organ development [8]. Type 2 EMT is associated with tissue regeneration and organ fibrosis [9]. Type 3 EMT is associated with cancer progression and metastasis and is the critical mechanism for the acquisition of malignant phenotypes by cancer cells [6]. Type 3 EMT promotes fibroblastoid morphology, cell migration and invasion, resistance to apoptosis and chemotherapy, and helps to maintain the overall cancer stem cell phenotype [10]. Molecularly, the cells undergo changes in gene and specific microRNAs expression, function, and/or activation of various proteins involved with this transition [3]. At the biochemical level, a metabolic reprogramming occurs. This is characterized by upregulated glycolysis, lipid metabolism, activation of the pentose phosphate pathway, and mitochondrial biogenesis, resulting in high energy production [11]. EMT also modifies the adhesive properties of the cells which leads to loss of the apical–basal polarity. Once EMT is triggered, the ability to produce and maintain cell–cell junctions is severely impacted. This decreased adhesiveness is due to a downregulation of canonical epithelial markers, such as E-cadherin, cytokeratins, and ZO-1 [6]. Furthermore, an enhanced expression of mesenchymal markers (N-cadherin, vimentin, fibronectin, α-SMA), the production of ECM-degrading enzymes, and ultimately the reorganization of the cytoskeleton and activation of transcription factors (TFs) that regulate EMT markers (Twist, Snail, Slug, ZEB, and β-catenin), are critical features of the trans-differentiation process [12]. 

### Role of Epithelial–Mesenchymal Transition in Cancer

In addition to its participation in normal physiological processes, EMT can be triggered by many oncogenic signaling pathways, hypoxia, and reactive oxygen species-mediated injury and signals of tumor microenvironment [13]. This results in a loss of cell polarity and adhesion and a gain of migratory and invasive properties to intravasate into the bloodstream, promoting metastasis in distant organs [4,5]. Moreover, tumor cells presenting with partial EMT already present increased invasive properties, which can generate circulating tumor cells and cancer stem cells, in addition to enhanced resistance to anti-cancer drugs [14]. 

During EMT, several signaling pathways are activated. Among these are the TGF-β, Wnt, Hedgehog (Hh), β-catenin, Notch, Nanog, and STAT3, as well as miRNAs and cytokines [6]. During the last few years, extensive research has established that additional signaling pathways also contribute to the activation and development of the EMT program; among these, the signaling pathway of ERK has been associated.

## 3. Extracellular-Signal Regulated Kinase: Structure and Function

The ERK pathway comprises several proteins that belongs to the family of mitogen-activated protein kinases (MAPK). These regulate diverse physiological and cellular processes, such as gene expression, cell division, survival, apoptosis, differentiation, and motility [15,16]. In metastatic processes, the MAPKs regulate EMT, invasion, angiogenesis, and resistance to therapy [7]. The MAPK family includes three subfamilies, which consist of 13 members divided into conventional and atypical kinases in humans. Within the conventional MAPKs, the following subfamilies are categorized: ERKs, c-Jun N-terminal kinases/stress-activated protein kinases (JNKs/SAPKs) and p38, and ERK1/2 and ERK5. ERK3/4 and ERK7/8 are atypical MAPKs [15,17]. Several members of the ERK subfamily participate in regulation of growth, differentiation, survival, and cell motility [18]. The JNKs and p38s subfamilies are involved in response to cellular events such as inflammation, infection, and environmental stress [19]. Six members of the ERK subfamily have been identified (Table 1, Figure 1): ERK1/2, 3/4, 5 and 7/8. Of these, only ERK3/4 do not present isoforms [15]. ERK1/2 are the two most studied kinases and the main regulators of the MAPK/ERK signal pathway [15]. In the following sections, we describe each of them in detail.

### 3.1. Typical Members of the ERK Subfamily

#### 3.1.1. ERK1/2

The genes that code for ERK1 and ERK2 are located on *16p11.2* and *22q11.21*; they have 10 and 9 exons, and molecular weights of 44 and 42 kDa, respectively [20]. ERK1/2 present a small amino-terminal and a large C-terminus [21]. The C-terminal domain is constituted by α-helical structures and contains six conserved segments (αD–αI). Four additional β chains (β6–9) contain the catalytic residues that participate in the transfer of phosphoryl groups from ATP to substrates of ERK1/2 [15,22]. ERK1/2 are activated mainly through cell surface receptors, such as the tyrosine kinase receptors (TKR) [21]. Upon ligand binding, the TKRs dimerize and the intracellular catalytic domain is transactivated, transducing the downstream signal by interacting with proteins that contain Src homology 2 (SH2) or phosphotyrosine-binding (PTB) domains [23]. These effectors activate subsequently the MEK1 and MEK2 kinases, which in turn phosphorylate ERK1/2 on Thr and Tyr residues [21]. Over 200 cytoplasmic, nuclear, membranous, or cytoskeleton substrates have been identified for ERK1/2 (discussed below) [22]. 

Human ERK1 and ERK2 share 84% identity and contain the Thr-Glu-Tyr motif in their activation loop [20]. Among the few differences between these kinases is that ERK1 presents a slightly longer N-terminal domain (17 amino acids) and contains two more residues in its C-terminal domain compared to ERK2 [24]. Despite the similarities of these kinases, murine models for *erk1* and *erk2* demonstrated different biological effects. For instance, while the *erk1*^−/−^ mouse is viable and fertile, the *erk2*^−/−^ model has embryonic lethality after the implantation stage [25]. Despite the differences observed in developing murine embryos, emerging evidence suggests that ERK1 and ERK2 are indeed redundant in function [20]. The lethal phenotype observed in the *erk2*^−/−^ mouse is likely due to differential expression of the kinases, which contributes as a whole to the overall activity of the kinases. Importantly, in vitro studies show that the differential expression of these kinases is due to a stronger transcriptional response of the promoter of *erk2* compared to the *erk1* promoter [26]. Furthermore, *erk2* depletion can be rescued by inducing the overexpression of *erk1* by using a stronger promoter, such as the one for *β-actin* [27,28,29]. ERK1/2 can induce the expression of immediate early genes (*c-Fos* and *c-Jun*) and control gene expression in response to growth factors, hormones, and cytokines [30]. Once MEK1/2 phosphorylates ERK1/2, it translocates to the nucleus to phosphorylate different targets, such as ETS (E-twenty six) and AP-1 [31]. Phosphorylation of these TFs leads to the activation (and in some cases inhibition) of transcriptional domains and stabilization of additional proteins involved in gene regulation (Figure 2). The regulatory roles of ERK1/2 are not limited to transcription; these kinases can regulate protein translation, synaptic plasticity, cell migration, EMT, cellular metabolism, survival, and mitochondrial fission [32,33]. 

#### 3.1.2. ERK5

ERK5, a conventional MAPK, is encoded at the *17p11.2* genomic locus. ERK5 has a molecular weight of 100 kDa, which makes it two-fold larger compared to other MAPKs, thus, it is also known as big MAP kinase 1 (BMK1) [34]. The ERK5 N-terminal domain shares 50% identity with ERK1/2 and also presents the Thr-Glu-Tyr activation domain [22,35]. Unlike the rest of conventional ERKs, ERK5 contains a nuclear localization signal (NLS) and a transactivation domain at its C-terminal end [22,35]. ERK5 is expressed in high levels in the brain, thymus, and spleen. It is proposed to regulate embryonic and vascular system development, neuronal activity, and neuroplasticity, as well as cell survival and proliferation [22,36]. Mechanistically, ERK5 is phosphorylated in the Thr-Glu-Tyr residues in the kinase domain by the protein interaction module PB1 of MEK5 [37,38]. 

In turn, MEK5 is phosphorylated by two upstream kinases, MEKK2 and MEKK3, which also activate JNK and p38. Importantly, the signal pathway for ERK5 activation is not entirely described yet and reported results remain controversial [39]. For instance, research has shown that MEKK2 and ERK5 compete for binding to the PB1 module of MEK5 [38], but it has also been proposed as a scaffold for the MEKK2–MEK5–ERK5 complex [37]. ERK5 can induce immediate early genes [40] and may regulate cell proliferation via the EGF pathway during the G1/S transition [40] by activating the serum and glucocorticoid-inducible kinase (SGK) through S-phase entry [41] and by inducing cyclin D1 expression [42]. 

### 3.2. Atypical Members of the ERK Subfamily

#### 3.2.1. ERK3/4

The ERK3/4 kinases are atypical members of the MAPK family, encoded by the *Mapk6* (*15q21.2*) and *Mapk4* (*18q21.1-q21.2*) genes, respectively [15]. Both kinases present a single phosphorylation site in the activation loop (Ser-Glu-Gly) and share 73% of homology in the kinase domain [22,43]. Unlike conventional MAPKs, basal phosphorylation of ERK3/4 is detected in cells and does not change in response to mitogens [44]. The regulation of ERK3/4 activity is unclear, however, it is known that ERK3/4 can be phosphorylated by p21-activated kinases class 1 (PAK1, PAK2, and PAK3) [43]. The only substrate reported for ERK3/4 is MAPK-activated protein kinase 5 (MK5 or PRAK) [43]. 

#### 3.2.2. ERK7/8

ERK7/8 has recently become relevant, since experimental evidence has generated information about its biological role [45,46]. ERK7 was identified initially in rats and later the human ortholog was identified and named ERK8; however, to avoid confusion, the protein is called ERK7/8 in humans [15]. This kinase shares 45% identity with ERK1. It is considered an atypical MAPK because it presents an extended C-terminal domain with an NLS, which is not present in conventional MAPKs [22]. Three isoforms have been reported for ERK7/8: The canonical isoform 1 of 544 amino acids and the smaller isoforms 2 (ERK8δ) and 3 with 254 and 277 amino acids, respectively [15,22]. These isoforms include the classic Thr-Glu-Tyr residues in the kinase domain. ERK7/8 can be phosphorylated at Ser, Thr, and Tyr residues; however, the kinase is activated only by phosphorylation at Thr175 and Tyr177 within the Thr-Glu-Tyr domain [15]. ERK7/8 is involved in cell transformation and has been associated with gastric [47] and colon cancer [48], although it might be differentially expressed and function across various cell types [45]. It is considered an onco-kinase that phosphorylates proto-oncogene c-Jun at Ser63/Ser73 to promote tumorigenesis [47,48]. 

## 4. Signaling Pathways Driving ERK Activation: From Membrane Receptors to Substrates

### 4.1. Signal Transduction by MAPKs

ERK1/2 are the most important members of the MAPK signaling pathway (Figure 2). These are activated in response to diverse extracellular stimuli such as cellular stress, growth factors (PDGF, EGF, NGF, insulin), cytokines, and ECM components [22]. ERK1/2 activation can lead to the stimulation of TKRs such as Epidermal Growth Factor Receptor (EGFR), G protein coupled receptors (GPCRs) and integrins [49]. The canonical activation pathway of ERK1/2 involves the dimerization and autophosphorylation of TKRs at Tyr residues [22,50], which serve as binding sites for the recruitment of the Src homology and collagen (SHC) adapter protein and subsequently growth factor receptor-bound protein 2 (GRB2) [23]. SHC and GRB2 are adapter proteins with phosphotyrosine-binding (PTB) and SH2 domains; thus, SHC recruits GRB2 through its SH2 domain [23]. GRB2 then interacts with the guanine exchange factor Son of Sevenless (SOS) through its SH3 domain. The resulting GRB2–SOS complex is bound to the cell membrane, where the GTPase RAS is located, and can be then activated by SOS [50].

The RAS family of small GTPases involved in the ERK1/2 pathway activation includes the H-RAS, K-RAS, and N-RAS proteins [51]. In unstimulated and quiescent cells, RAS proteins are anchored to the plasma membrane in an inactive state as monomers bound to GDP (Guanosine diphosphate) [52]. Upon a given stimuli, SOS associates with the inactive monomers of RAS, which then dimerize and are activated by stimulating the exchange of GDP for GTP (Guanosine triphosphate) [50]. Active RAS-GTP undergoes conformational changes at the Switch I and II domains located in its lobe effector. This process exposes an interaction site for downstream substrates and effectors such as RAF, phospholipase C (PLC), PI3K, and Tiam1, among others, which are recruited to the cell membrane and/or activated [53]. 

The main RAS effectors are the MAP3K proteins of the RAF family: A-RAF, B-RAF, and C-RAF. These are Ser/Thr kinases that contain RAF binding domains at the N- and C-terminal kinase domains [54]. The interaction between RAF and GTP-bound RAS promotes its recruitment to the cell membrane, followed by RAF dimerization upon dephosphorylation of the Ser259 by PP2A, exposing the kinase domain. In addition, RAF proteins can be phosphorylated by various kinases, such as PKC, PAK, c-Src, MLK3, and others [23,50]. Thus, it is well established that phosphorylation and dephosphorylation events modulate RAF kinase activity and its interaction with other proteins [55]. Experimental evidence shows that in quiescent cells N-RAS is constitutively interacting with both C-RAF and PKCε as an ordered, silent signaling module [56]. B-RAF can be activated by phosphorylation at Thr598 and Ser601 by RAS, RAP, and MLK3 [57]. The regulation of activity of A-RAF is related to the Tyr296 within the N-domain, which is a major determinant of the low kinase activity of A-RAF. A-RAF has an isoform-specific hinge segment (IH) domain which contains seven putative phosphorylation sites, of which three, Ser257, Ser262, and Ser264 are essential for the activation of A-RAF [58].

The main substrates of RAF kinases are MEK1/2, which are the MAP2Ks that activate ERK1/2 [59]. RAF and other kinases phosphorylate MEK1/2 at Ser218 and Ser222, located in the activation loop, thereby inducing their activation [60]. MEK1/2 are cytoplasmic dual-specificity kinases that phosphorylate residues of Tyr and Ser/Thr [61]. MEK1/2 recognizes and bind to the inactive form of ERK1/2 and may act as cytoplasmic anchor or activator [62]. One of the proposed regulatory mechanisms for the activity of MEK1/2 as cytoplasmic anchor is related to three Lys residues located at the N-terminus [61]. These amino acids are in proximity with several positively charged residues called cytoplasmic retention sequences (CRS) [61]. The CRS is a fundamental component for the establishment of the MEK1/2- ERK1/2 interaction at the common docking (CD) domain of the latter kinase [61]. A priming phosphorylation of MEK1/2 at Ser298 facilitates the contact between these domains and promotes the activation of MEK1 by RAF [55,60].

Interestingly, the only known substrates of MEK1/2 are the ERK1/2 kinases. This specific interaction prevents the simultaneous activation of other proteins of the MAPK pathway [55]. Furthermore, the N-terminal domain of MEK1/2 encodes a Leu-rich nuclear export signal located at residues 32–44 [62]. This sequence is recognized by the CRM1 nuclear export protein, which is necessary for the constitutive cytosolic localization of the kinase and also prevents the nuclear translocation of ERK1/2 in the absence of mitogenic signals [50,62,63]. MEK1/2 recognizes and phosphorylates ERK1/2 at Thr-Glu-Tyr residues: Thr202/Tyr204 for ERK1 and Thr183/Tyr185 for ERK2 [64,65]. This phosphorylation produces a conformational change in the phosphorylation and the nucleotide binding pocket (P + 1), which activates ERK1/2 [55].

In addition to the lobe phosphorylation, ERK1/2 contains a NLS encoding a Ser-Pro-Ser sequence that must be phosphorylated to allow the kinase to translocate into the nucleus [66]. Two substrate binding sites, named as D-recruitment site (DRS) and F-site recruitment site (FRS) [61,67,68], are important domains for signal transduction triggered by ERK1/2 activation by MEK1/2 [61,67]. The DRS is composed of a CD domain; two Asp residues in the CD domain and a hydrophobic groove interact with positively charged hydrophobic residues located in the kinase-interacting motif (KIM) or docking-peptide (D-peptide) [61,67,68]. On the other hand, the FRS allows the interaction of ERK1/2 with its substrates through the F-site or docking site for ERK–FXF (DEF), which encodes a Phe-X-Phe motif [61,67,68,69]. These two contact domains allow ERK to interact with its substrates in different ways: 1) The substrate’s DEF domain binds to ERK near the catalytic cleft, or 2) through the KIM domain, which is located at the opposite side of the kinase [61,69,70,71]. 

### 4.2. Substrates of ERK1/2

A myriad of interacting partners and effectors of ERK1/2 have been characterized experimentally. ERK1/2 can exert its effect, either in the cytosol or in the nucleus, upon different stimuli. In the cytosol, the active form of ERK can phosphorylate other kinases, phosphatases (PLA2, RSK) [23], and components of the cytoskeleton, mitochondria, endosomes/lysosomes, and Golgi [72]. Moreover, once ERK1/2 is autophosphorylated or activated by casein kinase 2 (CK2) at Ser244/246, it can interact with the nuclear importer protein importin 7 [73]. These phosphorylations lead to ERK1/2 translocation to the nucleus, where it mainly phosphorylates TFs [22,63].

Several substrates of ERK and regulatory proteins contain D-peptide, KIM, F-site, and DEF sequences which allow diverse interactions and activation by ERK [74]. ERK phosphorylates the conserved Ser/Thr-Pro residues and most of the substrates usually present another Pro at position –2 of the phosphorylation site (Pro-X-Ser/Thr-Pro) [55]. 

#### 4.2.1. Cytosolic Substrates

Among the main cytoplasmic substrates of ERK are the ribosomal S6 kinase (RSK), MAPK-interacting kinase (MNK), and mitogen and stress-activated protein kinase (MSK) proteins [75,76]. The proteins of the RSK family are exclusively activated by ERK1/2, whereas MNK1 and MSK1 can also be activated by p38 [77]. The four isoforms of RSK (RSK 1–4), and MSK1/2 contain two protein kinase domains; the first is located at the N-terminal end (the N-terminal kinase domain (NTKD)) and the other at the C-terminal end (the C-terminal kinase domain (CTKD)) [78]. These domains are linked by a single polypeptide chain called the linker region. The CTKD is involved in the protein’s autophosphorylation (activation) of RSK, while the NTKD phosphorylates the substrates. The RSK proteins present a D-peptide or KIM domain and the consensus sequence (S/TP) in the RSK activation loop at the C-terminal end; these are necessary for the Pro-directed phosphorylation by ERK1/2 [79]. 

RSK has six known phosphorylation sites of significance. These are Ser221, Ser363, and Ser380 in NTKD, Thr359 in the linker region, and Thr573 and Ser732 in CTKD [79]. The Thr573 residue is followed by a Pro, so ERK is directed to this site and phosphorylates the Thr, promoting the activation of CTKD. ERK also phosphorylates the Ser363 and Thr359 residues. The activated CTKD catalyzes the phosphorylation of Ser380 to allow the binding of phosphoinositide-dependent kinase 1 (PDK-1) to a constitutively active protein-Ser/Thr kinase, which in turn phosphorylates NTKD in Ser221, leading to full activation of RSK [80,81]. The autophosphorylation of the Ser732/3 might be important for the dissociation of the RSK–ERK complex and inactivation by phosphatases [64]. RSK modulates the phosphorylation of other proteins such as the TFs cAMP response element binding (CREB), serum response factor (SRF), estrogen receptor α (ERα), nuclear factor κ-light-chain-enhancer of activated B cells (NF-κB), nuclear factor of activated T cells 3 (NFAT-3), and transcription initiation factor 1A (TIF1A). The ribosome-associated proteins, like ribosomal proteins S6 and eukaryotic initiation factor 4B (EIF4B), pro-apoptotic proteins Bcl-2-associated death promoter (BAD), death-associated protein kinase (DAPK), and cyclin-dependent kinase inhibitor 1B, p27^Kip1^ (CDKN1B), are also targets of RSK [78]. Some proteins located in the plasma membrane can also be substrates of ERK, including connexin 43, which downregulates gap junction communication [82], and myosin light chain kinase (MLCK), which regulates cell migration [83]. 

Additional cytoplasmic substrates of ERK1/2 are the protein tyrosine phosphatases (PTPs), receptor-type tyrosine-protein phosphatase R (PTP-SL) and Stratium-enriched phosphatase (STEP), which contain a Leu-X-X-Arg-Arg site followed by several basic residues (same as the MAPKAPKs) [84]. The inactive forms of both ERK and PTPs interact in the cytoplasm, but upon ERK activation and PTP phosphorylation, the complex dissociates [77]. In addition, ERK phosphorylates some cytoskeleton components involved in maintenance of cell morphology, adhesiveness, migration [85,86], and for the recruitment of ERK to its proper cellular location and targets [64]. Paxillin is a focal adhesion specific scaffold protein of the ERK1/2 cascade that can also be phosphorylated by this kinase [87,88]. Paxillin is constitutively associated with inactive MEK. Upon hepatocyte growth factor (HGF) stimulation, RAF1 and ERK1/2 are recruited by paxillin facilitating the activation of ERK [89]. Then, ERK1/2 phosphorylates paxillin at Ser81 which regulates FAK signaling to induce local cytoskeletal rearrangement and cell morphogenesis [90]. Paxillin is a multi-domain docking protein that also associates with Src, thus having a second mechanism of signal transduction dependent on ERK1/2 [64]. 

#### 4.2.2. Nuclear Substrates

The main nuclear substrates of ERK1/2 contribute to transcriptional regulation, chromatin remodeling, cell cycle progression, motility, and EMT [55]. Nuclear substrates of ERK1/2 include the ternary complex factor (TCF) family of TFs, one of the main regulators of the immediate early genes *c-Fos* and *c-Jun* [61]. Among these, Elk1, a member of the TCF subfamily of the E-twenty-six (Ets)-domain TF, present both the D-peptide and F-site to enable its interaction with the CD-domain and (FSR) of ERK1/2 [61,91]. Importantly, the phosphorylation state of Elk1 mediates its transcriptional activity [91]. As such, Elk1 presents a C-terminal transcriptional activation domain with multiple ERK1/2 core consensus phosphorylation sites: Ser324, Thr336, Ser383, Ser389, and Ser422 [61]. c-Fos is a transcription factor involved in cells proliferation and differentiation [92]. Elk1 is one of the main regulators for *c-Fos* expression. Upon Elk1 activation, c-Fos is expressed within minutes or hours. Moreover, ERK1/2 also contributes directly to the stability of c-Fos, as the phosphorylation at Ser32 prevents the degradation of this TF [61]. A similar mechanism allows the stabilization and function of other immediate early genes such as *c-Myc* and *Fra1* [93]. 

### 4.3. Activation of ERK1/2 Pathway by G Protein-Coupled Receptors

In addition to the activation of the canonical mechanism described for the ERK1/2 pathway, there are alternative mechanisms that involve the transactivation of TKRs through G protein-coupled receptors (GPCRs) (Figure 3) [94,95]. Overall, the GPCRs lack intrinsic kinase activity, thus G proteins present a coupled function as signal transducers [96]. G proteins are heterotrimeric complexes composed of the Gα (bound to GDP), Gβ, and Gγ subunits [97,98]. Once activated, the GPCR acts as a guanine nucleotide exchanger to convert Gα-GDP to Gα-GTP; this process leads to the dissociation of the heterotrimeric complex Gα/β/γ [98,99]. The active Gα and Gβγ subunits modulate diverse cellular responses through effector molecules and second messengers, which, in turn, depend on the subtypes of G proteins coupled to the receptors [97,99]. There are four subtypes of the subunit Gα: Gαq/11, Gαs, Gαi/o, and Gα12/13 [96,99]. Considering the wide variety of GPCRs, G protein subtypes, and different effectors, two mechanisms for ERK1/2 activation dependent of GPCRs have been proposed: (1) Direct TKR-ligand interactions, or (2) ligand-independent process [49]. Moreover, the Gαq, Gαs, and Gαi subunits also activate proteins of the MAPK-ERK pathway to regulate cell proliferation, apoptosis, and survival [49]. 

### 4.4. Regulation of the ERK1/2 Pathway

Due to the variety of cellular processes and responses that the ERK pathway can trigger, it must be tightly regulated. The ERK1/2 regulatory systems involve cytoplasmic anchors, adapter proteins and phosphatases [100]. The cytoplasmic anchors maintain ERK1/2 in the cytosol until a mitogenic stimulus is detected, triggering the dimerization, activation, and nuclear translocation of ERK1/2 [101]. Inactive ERK1/2 also interacts with the actin cytoskeleton via the scaffold protein IQ motif-containing GTPase-activating protein 1 (IQGAP1) [102]. If stimulated, IQGAP1 recruits B-RAF and MEK1/2 to further activate ERK1/2 [103]. However, MEK1/2 remains as the main cytoplasmic anchor (discussed above) [62]. The MAP kinase phosphatase 3 (MKP-3) is another example that anchors ERK1/2 to the membrane and also dephosphorylates and inactivates the kinase [104,105]. 

The kinase suppressor of RAS (KSR) is another important adapter protein for ERK1/2-dependent signal transduction. KSR is a scaffolder that increases the signaling efficiency of the pathway by anchoring kinases such as B-RAF, RAF-1, MEK1/2, and ERK1/2, to the cell membrane [100,106]. The phosphoprotein enriched in astrocytes 15 (PEA-15) is another regulatory adapter protein of ERK2. PEA-15 activates the ERK pathway in a RAS-dependent manner by binding solely to ERK2 [100,107]. The C-terminal domain of PEA-15 interacts with an inverted D-peptide sequence in ERK2 to recruit the kinase to the cytoplasm. Moreover, an inhibitory effect of PEA-15 on ERK-dependent transcription and proliferation has been postulated to prevent its nuclear translocation [108]. The N-terminus of PEA-15 encodes for a nuclear export sequence and mediates the relocation of ERK and p-ERK from the nucleus to the cytoplasm in a manner similar to the MEK-dependent nuclear export of ERK [62,109]. A death effector domain (DED) in PEA-15 interacts with the FSR binding site in ERK2. Death motifs are known for their ability to recognize matching motifs in proteins that regulate apoptosis [100,107]. Experimental evidence showing an interaction between PEA-15 DED and ERK1/2 suggests a novel regulatory role for the DED motif [50,108]. 

Dual-specificity MAPK phosphatases (MKPs/DUSPs) also regulate the ERK signaling pathway. MKP3/DUSP6 can dephosphorylate ERK1/2 in both Tyr/Thr residues to inactivate it [110]. The nucleus is a critical site for mitogenic signal termination. DUSP5 is an anchor protein and an ERK-specific inhibitor, which contributes to nuclear sequestration of p42/p44 and prevents their interaction with MEK and subsequent dephosphorylation by specific nuclear phosphatases. This therefore determines the duration of the signal [111,112,113]. 

ERK5 is activated in response to serum growth factors, such as EGF, NGF, and hyperosmolarity [40,114,115], and more so by H_2_O_2_, suggesting that ERK5 is a redox-sensitive kinase [116]. ERK5 activation involves the Lys deficient protein kinase 1 (WNK1), which was identified as the potential upstream kinase for MEKK2/3. ERK5 activation also involves c-Src and the SH2-containing protein Lad, however, the mechanisms of action are poorly understood [117,118]. It has been hypothesized that the activation of a Tyr protein kinase such as Src [116] transmits the signal to the adapter protein Lad1 [118] and/or to WNK1 [117]. These in turn may phosphorylate and sequentially activate MAP4K, MEKK2/3, and finally MEK5 [117,119,120], which might be the upstream kinase for ERK5 [121]. 

Upon activation, ERK5 translocates to the nucleus and phosphorylates diverse substrates including some TFs, including the myocyte enhancer factor 2 (MEF2), Sap1a (ETS domain transcription factors), c-Myc, serum and glucocorticoid-activated kinase (SGK), connexin 43, and BAD [41,114]. All these downstream effectors are well characterized regulators of cell proliferation and survival [122]. Importantly, MEKK2/3 phosphorylates and activates MEK5. However, this is not the sole substrate; MEKK2/3 can also activate JNK and p38 [22]. ERK5 is part of a distinct MAP-kinase signaling pathway that is required for EGF-induced cell proliferation and cell cycle progression [40]. 

### 4.5. Activation of Pathways Dependent on Atypical ERK Members

The mechanisms of activation and biological effects of the atypical members of ERK3/4 and ERK7/8 are poorly understood [15,22]. The activation loop of ERK3/4 does not contain the classic Thr-Gly-Tyr motif. It is instead activated by phosphorylation of a Ser residue in the activation loop sequence Ser-Glu-Gly [22]. In vitro studies have shown the autophosphorylation ERK3 at Ser189 [123]. Moreover, an upstream MAP2K for ERK3/4 has also been proposed [44,123]. The only ERK3/4 substrate identified is the MAPK-activated protein kinase MK5 [124,125,126,127]. Also, the inducible nuclear dual specificity MAP kinase phosphatase DUSP2 allows the dephosphorylation of both ERK3 and ERK4. This interaction is mediated by a conserved CD domain within the C-terminal domains of ERK3 and ERK4 and the conserved KIM domain located within the non-catalytic N-terminus of DUSP2. DUSP2-mediated dephosphorylation of ERK3/4 inhibits the activation of the downstream substrate MK5 [128]. Although the biological role of ERK4 is well not defined, ERK3 has been associated with proliferation, cell cycle progression, and differentiation processes [42,129,130]. 

ERK7/8 phosphorylation occurs upon stimuli like serum, estrogens [22], glucocorticosteroids [131], and an oxidative environment such as H_2_O_2_ stress [132,133]. The activation loop of ERK7/8 contains the conserved Thr-Gly-Tyr motif; however, the precise mechanism of activation is not known. It was reported that the Thr-Gly-Tyr motif of ERK7 is constitutively phosphorylated [125]. Evidence suggests that autophosphorylation is key for its regulation. In this regard, it has been proposed that the atypical C-terminal extension of ERK7 is required for self-activation and also determines its subcellular location [45,125]. On the other hand, the expression of an oncogenic Src allele promotes the phosphorylation of the Thr-Gly-Tyr motif of ERK7/8. Src is activated following serum stimulation and does not require of the activity of MEK1/2 or MEK5 [132]. This evidence suggests the potential existence of an uncharacterized MAP2K involved in the activation of the human ERK7/8 [22]. 

The regulation of ERK8 activity in mammalian cells is determined by the relative rates of ERK8 autophosphorylation at Ser126 and its dephosphorylation at Thr175 catalyzed by PP2A or PTP1B [133]. Some of the substrates identified in vitro for ERK7 are the immediate early genes *c-Fos* and *c-Myc*. A common substrate for ERK7 and ERK7/8 is the myelin basic protein (MBP) [45]. ERK8 is also a negative regulator of human GRα, acting through Hic-5, which suggests a broader role for ERK8 in the regulation of nuclear receptors beyond the ERα [46]. Despite the limited knowledge of the specific substrates of these kinases, several studies suggest the possibility that ERK7 contributes to the inhibition of growth under stress [45,46,131]. 

## 5. ERK-Activating Mutations in the MAPK Signaling Pathway and Their Contributions to EMT and Cancer Progression 

In the last few decades, several mutations have been identified in molecules involved in MAPKs signaling pathway. Overall, these mutations lead to the hyperactivation of ERK1/2, and are associated with tumor progression. 

The mutations in the small GTPase RAS were the first oncogenic mutations described in a murine model of leukemia more than 30 years ago [134,135]. Mutations in the *Ras* gene occur in approximately 15%–30% of cancers [135,136,137]. According to the Catalog of Somatic Mutations in Cancer (COSMIC), 27% of RAS mutations are missense substitutions and 98% are hotspot mutations [138]. The most frequent hotspot mutations in cancer are G12 (83%), G13 (14%), and Q61 (2%) [138]. RAS mutations in T24 bladder cancer cells, pancreatic cancer MIA PaCa-2, rhabdomyosarcoma RD, and MCF10A mammary epithelial cells prevent GTP hydrolysis mediated by GTPase-activating proteins (GAP), which results in an accumulation of GTP-bound RAS and hyperactivation of RAS [138,139,140]. Mutations in codon 12 (*H-Ras*), 13 (*K-Ras*), and 61 (*N-Ras*) are oncogenic as well [141,142]. Also, it was estimated that K-RAS mutations predominate (85%) in cancer compared to H-RAS (4%) or N-RAS mutations (11%) [138].

It is estimated that in the Ser/Thr kinase *B-Raf* gene, approximately 30 missense mutations in the kinase domain are oncogenic [143]. Particularly, in A549 lung cancer cells, analyses of the mutants ERK3-KD and ERK3-Ser189Ala showed that the phosphorylation of Ser189 is necessary for MK5 activation [144]. Mutant ERK3-KD binds ATP, while mutant ERK3-Ser189Ala cannot [144]. Interestingly, the ERK3-KD mutation decreased the migration ability of A549 cells; however, the effect is more dramatic in cells with the ERK3-Ser189Ala mutation [144]. On the other hand, in vitro mutagenesis studies of ERK4 in HeLa cells (Asp168Ala and Ser186Ala) showed that both residues are essential for the activation of ERK4 and further, that Ser186 is required for stabilization of the complex ERK4-MK5 and activation and localization of MK5 [128]. 

A common mutation in B-RAF is the substitution Val600Glu, which occurs in at least 7% of tumors [145]. This mutation mimics the conformational changes mediated by phosphorylation of GTP-bound RAS at Thr599 and Ser602 in the activation segment of the kinase domain. Under physiological conditions the phosphorylation of Thr599 and Ser602 is key for the activation of B-RAF [143,146]. Dimerization is an important regulator in ERK signaling [147]. Dimerization is essential for the normal function of B-RAF. The Val600Glu mutation in B-RAF promotes the monomer form with an increased kinase activity [148,149]. About 40% of individuals with melanoma have mutated B-RAF, which modifies its structure and/or function [150]. Interestingly, mutations in MEK1/2 have been found in patients with melanoma, lung, and colorectal cancer [27,151]. This leads to an additional regulatory feedback loop upstream of MEK1/2 provided by B-RAF; the single base mutation Val600Glu renders a constitutively active enzyme which constantly phosphorylates MEK1/2 [148,149]. Likewise, in patients with multiple myeloma, the B-RAF Val600Glu mutation is also associated with an increase in ERK activity [152]. In vitro studies in MCF10A cells showed that the Val600Glu mutation favors EMT by inducing morphological changes associated with impaired cell-–ell junctions, decreased E-cadherin expression, and increased vimentin levels [153]. Conversely, in breast cancer, the AVKA (Thr599Ala/Ser602Ala) mutation prevents the phosphorylation of Thr599 and Ser602 of B-RAF, which enhances the invasive capacity and metastasis potential of the tumor cells [154]. B-RAF Glu586Lys is another mutation involved in tumor progression; it increases the dimerization potential of B-RAF [149,155]. Furthermore, studies in NIH 3T3 cells showed that the Ser729 mutation in B-RAF suppresses its kinase activity [156]. 

A study of patients with multiple myeloma showed that 24% of the individuals have mutations in N-RAS, 25% in K-RAS, and 9% in B-RAF [152]. In most cases, RAS/RAF mutations were mutually exclusive (90.6%) [152]. Likewise, the codons affected in the case of B-RAF were in codon 600, in N-RAS they were in codon 61, and in KRAS, codon 12. The most frequently occurring mutations were in K-RAS codon 12. The most frequently occurring mutations for K-RAS were Gln61His, Gly12Asp/Gly12Val, and Gly12Ala, for N-RAS they were Gln61Arg, Gln61Lys, Gly13Asp, and Gly13Arg/Gln61His, and as expected, the most common mutation in B-RAF was the Val600Glu mutation [152]. Additionally, in melanoma, colorectal, and thyroid cancer, 18% of the patients had mutations in KRAS; of which 11% were comprised of the Gly12Ala and Gly12Cys mutations, 32% were Gly12Asp, 28% were Gly12Val, 12% were Gly12Ser, and 17% were Gly13Asp [136]. 14% of patients had the B-RAF Val600Glu mutation [136]. Likewise, in a study conducted in pilocytic astrocytoma patients, it was observed that 9.38% of individuals presented with the Val600Glu mutation in B-RAF [157]. 

Additional mutations in ERK1/2 have been reported in several types of cancer and cell lines. For instance, the in vitro models A375, 293T, SKMEL-19, and WM266.4 cell lines showed that the mutants ERK1 Tyr53His/ERK2 Tyr36His/Asn, ERK1 Gly54Ala/ERK2 Gly37Ser, ERK1 Pro75Lys/ERK2 Pro58Lys, ERK1 Tyr81Cys/ ERK2 Tyr64Asn, and ERK1 Cys82Tyr/ERK2 Cys65Tyr had resistance to ERK inhibitors, favoring tumor progression [158]. On the other hand, the mutants for ERK1 Ile48Asn, Tyr53His, Gly54Ala, Ser74Gly, Pro75Lys, Tyr81Cys, Cys82Tyr, and Gly186Asp and for ERK2 Tyr36N/His, Gly37Ser, Pro58Lys, Tyr64Asn, and Cys65Tyr interfere with the coupling of ATP/ERK1/2 with their inhibitors [158]. However, the ERK1/2 mutant Gly54Ala and ERK2 Gly37Ser are rendered as enzymes with lower kinase activity compared to wild type [158]. Furthermore, in vitro studies using various cancer cell types, such as A375, IPC298, SKMEL30, HCT116, MIA, PaCa2, and Panc1, pointed to several additional mutations that cause resistance to ERK inhibitors and favor EMT and tumor progression. Relevant mutations for ERK1 include Tyr36His, Cys65Phe/Tyr, Gly37Ala/Ser/Cys, Ala191Val, and Gly186Asp and for ERK2, Tyr36Asn and Gly37Ser [159]. While in HCT116 and SKMEL30 cells, the mutants of ERK1 Cys65Phe, Tyr36His/ ERK2 Gly37Cys, and Gly37Ala conferred a similar effect [159]. 

Overall, the mutations described for the different proteins of the MAPK pathway ultimately lead to the phosphorylation of ERK. However, several studies in patients have elucidated the relationship between the phosphorylation of ERK and tumor progression. These findings are described in the following sections.

### p-ERK as a Marker of Cancer

During EMT, various proteins associated with diverse signaling pathways are post-translationally modified. The most relevant are MAPKs, tyrosine kinase, the cadherin–catenin complex, and cyclin-dependent kinase. These are the major players of the cell cycle, thus deregulation of their phosphorylation–dephosphorylation state can contribute to cancer progression [160]. Furthermore, phosphorylation of effector proteins, like the TFs Snail, Twist, and ZEB, also contributes to EMT and cancer progression [161]. 

To study the frequency of ERK activation in primary liver cancers, p-ERK was analyzed using a tissue microarray (TMA) containing 78 hepatocellular carcinoma specimens by immunochemical analysis. Levels of p-ERK in different tumor tissues were classified as high expression in 11 cases (11/78, 14.1%), low expression in 11 cases (11/78, 14.1%), or negative in 56 cases (56/78, 71.8%) and p-ERK was detected in both the cytoplasm and nucleus [162]. Previous reports show that the phosphorylation levels of RAF, MEK, and ERK were significantly upregulated in samples from laryngeal cancer patients and contributed to tumor progression [163]. On the other hand, SHP2 promotes laryngeal cancer growth through the RAS/RAF/MEK/ERK pathway and serves as a prognostic indicator for laryngeal cancer [164]. In hepatocellular carcinoma patients, p-ERK was localized to the nucleus of tumor cells and higher levels of p-ERK were associated with increased progression time. These findings suggest that MAPK signaling is a significant and clinically relevant predictor of treatment outcome [165]. ERK is also constitutively active in ovarian cancer. Elevated levels of p-ERK were identified in histological sections of high-grade ovarian cancer. Elevated p-ERK levels were tightly correlated with cancer progression, especially in high-grade ovarian cancer [166]. All these results strongly suggest that the phosphorylation of ERK is related to the activation of the EMT program.

## 6. Relationship between ERK and Epithelial–Mesenchymal Transition in Various Cancer Types

In this section, we highlight the experimental evidence obtained from different cancer cell lines, animal models, and biopsies of patients, which have established the relationship between ERK and EMT in different types of cancer (Figure 4).

### 6.1. Lung Cancer

A549 lung cancer cells and MLCC mesenchymal lung cancer cells stimulated with TGFβ showed increased levels of p-ERK1/2. This correlated with enhanced expression of β3 integrin, vimentin, fibronectin, Slug, MMP-9, and decreased levels of E-cadherin [167]. In the lung cell line A549, ERK1/2 also promoted the expression of *Zeb1* through Fra1 [168]. Overexpression of ZEB1 decreased the expression of E-cadherin and increased the expression of fibronectin [168]. In the mesenchymal lung cancer cells TD, the MAPK and PI3K/AKT pathways were also hyperactivated [169]. In the H1299 lung cancer cell line, ERK3 phosphorylated the Ser857 of the co-activator of the steroidal receptor SRC-3, which is necessary for its interaction with the TF PEA3. This complex promotes the overexpression of MMP-2, MMP-9, and MMP-10 to increase cell migration and invasion [170]. In lung cancer, leptin is also known to induce EMT-associated morphological changes by decreasing expression of E-cadherin and keratin and increasing expression of the mesenchymal markers vimentin and ZEB1. This further activates the ERK signaling pathway [171].

In studies using lung tissue from BALB/c mice exposed to tobacco, a decrease of E-cadherin and ZO-1 and an increase of N-cadherin and vimentin were found to induce EMT. Tobacco also increases the activity of ERK1/2 while decreasing the activity of ERK5 [172]. Although the role of ERK5 in the regulation of EMT is not yet completely understood, the *Erk5* knockdown in A549 cells resulted in a decreased expression of Snail and ZEB1, both transcriptional repressors of CDH1 [173]. An additional study using a model of pemetrexed-resistant lung cancer cells showed that ERK1/2 contributed to the induction of ZEB1 and fibronectin while decreasing the expression of E-cadherin, therefore inducing EMT [168]. 

### 6.2. Breast Cancer

TGF-β is well-known as a classical inducer of EMT in breast cancer. Studies using the highly invasive triple-negative cell line MDA-MB-231 showed that inhibition of TGF-β blocks the ERK/NF-kB/Snail signaling cascade and contributes to the recovery of the epithelial phenotype while decreasing cell migration [174]. In the breast cancer cell lines MCF7 and T47D, the EMT is favored by a positive feedback mechanism between ERK1/2 and Twist, which stabilize and enhance function of both proteins [175]. In addition, the phosphorylation at Ser68 of Twist is essential in promoting EMT and invasion in breast cancer cells, although it does not diminish the expression of E-cadherin. It instead contributes to the expression of vimentin [176]. Additional evidence that establishes the role of ERK in EMT was obtained from the non-tumoral epithelial breast cells MCF10A, where the ectopic expression of the Scribble protein, which is essential for the maintenance of apical–basal polarity, decreased the phosphorylation state of MERK1/2 and ERK1/2 [177]. By decreasing the phosphorylation of ERK1/2, the expression of FRA1 also decreases and consequently ZEB1 decreases as well. This leads to an increase in the expression of ZO-1, claudin, and occludin and adequate formation of adherent junctions [177]. Likewise, the expression of ZNF259 (Zinc finger protein 259) in MCF-7 and MDA-MB-231 cell lines increases the phosphorylation of MEK, ERK, and GSK3β and expression of Snail, while diminishing the ERK-dependent expression of EMT markers [16]. On the other hand, in the T47D and MDA-MB-231 cell lines, a positive feedback was observed between the EMT transcription factor Snail and p-ERK, which favors the expression and nuclear localization of both proteins [178]. In these cells, the knockdown of Snail decreases the levels of p-ERK and its relocation to the cytoplasm, which in turn decreases the levels of Snail to ultimately allow an increase of E-cadherin and decreased vimentin [178]. Both the ERK1 and the ERK2 isoforms regulate the expression of Snail, but interestingly the ERK2 isoform has a greater effect on the regulation of Snail, suggesting that ERK2 may be the primary regulator of Snail-mediated EMT when compared to ERK1 [178]. In murine mammary epithelial cells (NMuMG), ERK1/2 is also known to mediate the global epigenetic reprogramming during EMT. Changes in transcriptional profiles in epithelial and mesenchymal genes whose promoters and enhancers are accessible and marked by acetylation of Lys27 at Histone 3 are influenced by ERK signaling [179]. Also, our group demonstrated that leptin induces partial-type EMT in MCF10A cells, regulating vimentin and cell migration in an ERK-dependent pathway [180]. In the breast cancer cell lines MCF-7/AKR1B10 and BT-20, the MEK1/2 inhibitor PD98059 attenuated the activation of ERK and expression of MMP-2 and vimentin, thus suppressing migration and invasion [181]. An interesting study in primary breast tumor samples showed that EpCAM promotes ERK activity, therefore elucidating novel double-negative feedback between EpCAM and ERK that contributes to EMT regulation [182]. In biopsies of patients with breast cancer and high-grade tumors, increased levels of p-ERK1/2 have been detected. This is accompanied by a nuclear increase in the expression of Snail [178]. 

Studies of other members of the ERK family have shown that in MDA-MB-231 and MCF7 cell lines, the overexpression of ERK3 induces changes in the actin cytoskeleton, decreasing the stress fibers in the periphery and promoting cell movement [183]. Interestingly, the knockdown of ERK5 in the breast cancer cell line MDA-MB-231 and other in vitro cancer cell models leads to a upregulation in *CDH1*, a decrease of the mesenchymal marker *SERPINE1* (the plasminogen activator inhibitor), and an associated decrease in cell migration and invasion [173]. Furthermore, enhanced cytoplasmic expression of ERK5 was observed in biopsies obtained from breast cancer patients, which was associated with a decrease in cell survival [184]. 

### 6.3. Colorectal Cancer

Regarding colorectal cancer, HGF induces EMT and an increase in p-ERK and AKT in the CT26 murine colorectal cancer cell line, that in turn regulates the expression of Slug [185]. The colon cancer cell lines HT-29, LS 174T, and SW620 express high levels of high-mobility group AT-hook 2 (HMGA2) transcriptional regulating factor, compared to the normal colon epithelial cell line CCD18Co. In HT-29 and COLO 205 cell lines, the elevated expression of HMGA2 was shown, like ERK1/2, to also be dependent of B-RAF [186]. Additionally, HMGA2 induces invasion and cell migration and regulates EMT markers in the colon cancer cell lines HT-29 and LS 174T. These changes are characterized by decreased expression of E-cadherin and ZO-1 and increased expression of vimentin, N-cadherin, and Slug [186]. 

A study performed on 160 biopsies of patients with colon cancer showed that 54% of the cases had increased levels of p-ERK and were strongly detected in cells located at the infiltrative tumor edge. Of the cases, 75% of the samples also had high expression of FRA1, which was also stronger in the infiltrative tumor cells and colocalized with p-ERK [187]. Also, the constitutive expression of active MEK in a mouse model of colon cancer induced an increase in the expression of Snail and β-catenin and also decreased levels of E-cadherin [187]. 

### 6.4. Cervical and Ovarian Cancer 

TGF-β was shown to induce EMT in the cervical cancer cell lines SiHa and C33A. In these cases, both the decrease of E-cadherin and increase of vimentin and fibronectin were regulated by CK17 (cytokeratin 17); in these models, CK17 is regulated by ERK1/2 [188]. On the other hand, in the human uterine HEC-1A cell line stimulated with recombinant Osteopontin, an increase in the phosphorylation levels of ERK1/2 and AKT were observed. Both of these pathways regulate the expression of MMP-2, vimentin, and N- cadherin, promoting EMT and invasion [189]. Studies in OVCAR3 and OVCAR5 ovarian cancer cell lines have reported that Rac1 is indispensable for the activation of the Src and ERK1/2 signaling pathways. Rac1 regulates EMT and cell migration by decreasing E-cadherin and increasing vimentin, ZEB1, ZEB2, and Twist expression [190]. In addition, a correlation between the decrease in the expression of the microRNA precursor miR-7 and the increase in metastasis invasive tissue samples and cell lines HO-8910pm, CAOV-3, SKOV3, and ES-2 was observed [191]. In ovarian cancer cell lines HO-8910PM and ES-2 transfected with the miR-7 plasmid, decreased phosphorylation levels of AKT in Ser473 and ERK1/2 in Thr202/Tyr204, as well as decreased levels of vimentin and increased levels of cytokeratin 18 and β-catenin suggest that miR-7 reversed EMT through the inhibition of AKT and ERK1/2 [191]. 

### 6.5. Prostate Cancer

ERK1/2 signaling is also involved in prostate cancer via different activation pathways. For instance, in vitro studies using the DU145 prostate cancer cells showed that the knockdown of ZEB1 decreases the phosphorylation of ERK1/2, without affecting the expression of total ERK1/2 [192]. This inhibition promotes apoptosis, inhibits cell proliferation, and restrains cell invasion [192]. A representative example of an effector upstream of ERK1/2 is Polo-like kinase 1 (PLK1). Ectopic expression of PLK1 in the cultured cell model RWPE-1 showed an increase in the phosphorylation of C-RAF, which subsequently activated MEK1/2 ERK1/2 [193]. ERK1/2 regulates the expression of ZEB1 and ZEB2 through FRA1 and therefore also regulates EMT by decreasing the expression of E-cadherin and cytokeratin 19 and increasing the expression of N-cadherin, vimentin, and fibronectin [193].

### 6.6. Other Types of Cancer

In hepatocellular carcinoma (HCC) cells cultured in medium conditioned with cells isolated from hepatocellular carcinoma patients [194], an increase in p-ERK1/2 levels was observed. As expected, elevated p-ERK1/2 promoted the expression of EMT markers, inducing a decrease in the levels of expression of E-cadherin and an increase in the levels of vimentin, Snail, and Twist [195]. Biopsies of patients with hepatocellular carcinoma revealed elevated levels of p-ERK1/2, which are largely located to the nucleus, compared to adjacent healthy tissues. Additionally, the levels of p-ERK1/2 correlate positively with vascular invasion and α-SMA expression [195].

In human osteosarcoma U2OS cells, TGF-β1 induces an increase in AKT and ERK1/2 phosphorylation. Both phosphorylation pathways decrease E-cadherin levels and increase the levels of vimentin, Snail, and Slug [196]. Additionally, in osteosarcoma MG63 and M10 cells cysteine-rich angiogenic inducer 61 (Cyr61) promotes EMT through the αvβ5/RAF-1/MEK/ERK pathway and induces a decrease in E-cadherin and an increase in the expression of N-cadherin and Twist [197]. TGFβ was also detected in the pancreatic cancer cell line Capan-1M9. In this case, ERK1/2 regulates the expression of Slug, N-cadherin, and CD133, the latter being a marker of cancer stem cells [198].

In melanoma cell lines generated from invasive and pre-metastatic tumors, the EMT-associated TF Twist is commonly overexpressed. Interestingly, Twist overexpression is enhanced in cell lines with B-RAF mutations [199]. Likewise, it was demonstrated that both ERK1/2 signaling is indispensable for the regulation of Twist at the mRNA and protein level [199].

T24 bladder cancer cells treated with benzidine, a cancer inducer, present a significant increase in p-ERK1/2 and p-ERK5, while in EJ bladder cancer cells, only an increase in p-ERK5 was observed [200]. In addition, by inhibiting the activity and expression of ERK5 in T24 cells, the levels of expression of E-cadherin and ZO-1 were restored and the levels of vimentin, N-cadherin, and Snail decreased; therefore, migration and cellular invasion also decreased [200]. However, by inhibiting ERK1/2, mesenchymal alterations in T24 cells were not reversed [200]. Thus, ERK5 plays an important role in the induction of EMT by benzidine in T24 cells in this cellular model, while ERK1/2 do not.

The knockdown of caveolin 1 (Cav-1) in the mouse peritoneal mesothelial cell line MC induces a decrease in E-cadherin and ZO-1 and an increase in the levels of α-SMA, vimentin, Snail, and also in the activity of ERK1/2. In this model, ERK inhibition with CI-1040 reduces Snail expression, allowing the expression of E-cadherin, ZO-1, and Cav1. The morphology to an epithelial phenotype is also recovered [201]. 

In gastric cancer, miR-592 induces EMT and promotes its progression via PI3K/AKT and MAPK/ERK signaling pathways by directly binding to 3′UTR of Spry2 and triggering its RNA degradation, thus modifying E-cadherin, vimentin, and N-cadherin expression, which are hallmarks of EMT [202]. 

The EMT in glioma tissues is correlated with high expression of ring finger protein 138 (RNF138), which regulates caspase 3, E-cadherin, p-ERK1/2, vimentin, MPP-2, HIF-1α, and VEGF [203]. In this case, ERK phosphorylates MCRIP1, releasing the co-repressor CTBP. Then, ZEB1 is enabled to bind to the promoter of target genes such as E-cadherin, whose regulation is a hallmark of EMT by stabilizing the adherent junctions [204]. In glioma tissue, ERK5 is also overexpressed, however the mechanism of action is not fully understood [205]. 

## 7. ERK as a Therapeutic Target in Cancer

To date, designing therapeutic targets for the RAS-RAF-MEK-ERK1/2 signaling cascade have been focused on RAF and MEK; some of these inhibitors are approved and largely used in clinical therapies [27,28,29]. However, some patients may develop innate resistance to these drugs, as re-activation of the ERK1/2 cascade relies on negative feedback and several other mechanisms [27,206,207]. Therefore, it is necessary to design innovative therapeutic strategies to target this signaling cascade at different levels. Complications from the selection of the appropriate drugs for each patient, and the potential for developing drug resistance, speak to the fundamental need to develop inhibitors for ERK1/2, the ultimate effectors of this signaling cascade. Thus, ERK1/2 inhibitors are of significant interest as a supplement to RAF and MEK therapies for those patients that have developed resistance to these drugs. Most of the ERK inhibitors developed to date are designed to target the kinase’s catalytic activity (type I) and some may interfere with the kinase’s phosphorylation by MEK1/2 at the pThr-Glu-pTyr motif. Covalent ERK inhibitors are also being designed to target the catalytic domain of the kinase. Additionally, allosteric inhibitors of ERK1/2 have been designed to restrain the kinase binding abilities with diverse interactors [29]. Although several small molecule inhibitors of ERK1/2 have been developed, few of them have advanced to clinical trials (Table 2). 

The small molecule FR180204 is amongst the first inhibitors shown to impair the kinase activity of ERK1/2 in vitro [208]. Structural and biochemical characterization of FR180204 binding to ERK2 demonstrated that this drug interacts with the ATP-binding pocket and competes with the nucleotide. FR180204 presents a high affinity and selectivity for ERK1/2 (IC_50_ of 0.51 μM, and 0.33 μM, respectively). This small molecule also inhibits the activity of the MAPK p38α, but at a higher IC_50_ (10 μM). At the cellular level, FR180204 decreased the phosphorylation of ERK1/2 target proteins like the myelin basic protein [208]. 

SCH772984 is based in a 3(S)-thiomethyl pyrrolidine structure and is a potent and highly selective inhibitor of ERK1/2. SCH772984 has an IC_50_ of 1 nM against ERK2 and 4 nM for ERK4. It also inhibits the activating phosphorylation of ERK by MEK (100 nM) and inhibits phosphorylation of the ERK downstream substrate RSK. SCH772984 is less effective against MEK1 (IC_50_ >10 µM) and at 1 µM, this small molecule partially inhibits CLK2, FLT4, VEGFR3, GSG2, MAP4K4, MINK1, PKCµ, and TTK. Further, prolonged inhibition of ERK with this compound inhibits C-RAF phosphorylation dependent on ERK at Ser289, Ser296, and Ser301 [209]. The structure of SCH772984 bound to ERK1/2 showed the formation of a binding pocket located between the kinase’s phosphate binding loop and the Tyr64, which leaves the DFG motif in an active conformation [214]. Since the binding site for SCH772984 is not a constitutive structure of the kinase, it is noteworthy that this chemical can bind to both the inactive and the active form of kinase. This drug suppresses cell proliferation in cancer cells that were resistant to a combination of B-RAF and MEK inhibitors. For instance, SCH772984 inhibits ERK and RSK phosphorylation (2 µM) in cases where the combination of PLX4032 and Trametinib (B-RAF and MEK inhibitors, respectively) failed to inhibit the signaling cascade. Unfortunately, the chemical properties of SCH772984, such as a high molecular weight, slow binding, and poor ligand efficiency, have restricted its development as a potential drug [214]. However, novel drugs have been developed based on SCH772984. One example is MK-8353/SCH900353 [215], which is a smaller molecule with higher ligand efficiency and comparable potency to SCH772984 in several preclinical cancer models [215]. Mechanistically, MK-8353 appears to act by a still unknown mechanism, different to the inhibition of ERK1/2 phosphorylation at the Thr-Glu-Tyr motif [210]. Thus, both compounds may be considered type I and II ERK1/2 inhibitors, since they inhibit ERK1/2 catalytic activity and also impair the kinase’s phosphorylation and activation by MEK1/2 at the Thr-Glu-Tyr motif [214,216]. SCH772984 and its derivate MK-8353/SCH900353 reached the clinical trial phase (ClinicalTrials.gov Identifier: NCT01358331). This study showed that MK-8353 was well tolerated by patients; however, it presented limited anti-tumor activity that did not correlate to the pharmacodynamic parameters observed in patients with B-RAF Val600 mutated melanoma [215]. 

A highly selective ATP-competitive inhibitor of ERK1/2 is VTX-11e. Cell-based assays showed that VTX-11e impairs the phosphorylation of the kinase’s substrate, p90RSK [208]. Novel drugs were designed based on the pyrrole structure of VTX-11 [217,218]. For instance, BVD-523, also named ulixertinib [211,219], has advanced to clinical trials in cases of advanced-stage malignant tumors, including B-RAF-mutated colorectal cancer, melanoma, non-small cell lung cancer, N-RAS-mutated melanoma, acute myelogenous leukemia, myelodysplastic syndromes, and MEK- and ERK-mutated cancers (NCT01781429, NCT02296242, NCT02608229). In vitro studies of ulixertinib reduced the proliferation and enhanced caspase activity in sensitive cells. This drug did not inhibit ERK phosphorylation, but impaired the phosphorylation of targets [219]. In xenografts, ulixertinib caused dose-dependent growth inhibition and regression of tumors. When combined with B-RAF inhibitors, ulixertinib prevented proliferation synergistically in a xenograft model using melanoma cell lines. This is a promising therapy that is still under clinical evaluation [219]. 

Tetrahydro-pyrazolopyridine-based ERK2 inhibitors continue to be developed. This chemical structure is known to improve potency and lipophilic efficiency of the drugs, which is achieved by a lipophilic aryl-Y interaction with the enzyme. Upon cleavage of the lipophilic moiety, the ligand-bound binding cleft undergoes reconfiguration to establish a polar contact between the released N-H and a neighboring Asp of the kinase, thus impairing its function [212,220]. Further, a novel series of ERK1/2 inhibitors based on pyrrolopyrazinone compounds were developed as ATP competitors, as potential anti-tumor drugs [221]. Studies in A375 melanoma cells showed that “Compound 16” (from AstraZeneca) blocks the activation of the kinase; chemical modification of the basic structure of these small compounds will continue to advance the design of novel anti-cancer drugs [221]. For instance, the pyrrolopyrazinone core is found in marine sponges [222]. Changes in chirality of methyl groups in the structure of these compounds have significant impact on the inhibitory properties against the activation and in the catalytic activity of ERK1/2 [221]. Importantly, chiral-modified pyrrolopyrazinone-based compounds have cytotoxic effects in vitro in different cancer cell lines, such as lung, leukemia, SKOV-3, A549, and HeLa [223,224]. Structure analyses showed that these compounds may interact with the Gly-rich loop on ERK1/2 which leads to inhibition of the kinase; however additional mechanistic characterization is required. Among these small molecule inhibitors, “Compound 35” (from Astra Zeneca) has an IC_50_ less than 0.3 nM against ERK1/2 in vitro and in A375 cells the IC_50_ was 12 nM for p-ERK and 60 nM for p-90RSK, suggesting a dual mechanism of action similar to that observed for VTX-11e [221]. This novel small molecule binds covalently to the ATP pocket in the kinase. Another derivative is “Compound 13”, which had an in vitro IC_50_ of 6.9 nM against ERK2 activity and an IC_50_ of 53 nM in an A375 cell proliferation assay [221]. Whether these drugs are adequate for treatment of cases where RAS is constitutively active remain to be elucidated in clinical trials. 

ERK inhibitors that impair the kinase’s catalytic activity induce the accumulation of phosphorylated ERK, which releases it from cytoplasmic MEK1/2 to be translocated to the nucleus [72,158,216]. Although these inhibitors might impair the kinase’s activity by contributing to the accumulation of inactive p-ERK1/2 in the nucleus, there is a strong potential for the enzyme to maintain its non-catalytic nuclear function of ERK1/2 [29]. In this scenario, these inhibitors have a poor response in cellular systems. Therefore, dual inhibitors, with the capacity of impairing the catalytic activity and the phosphorylation of ERK, represent the best alternative to drugs with only one target activity.

Covalent ERK1/2 inhibitors have been designed to target signaling pathways in a sustained fashion. These molecules interact with their targets by initially establishing a reversible electrophilic interaction facing toward a nucleophilic residue to form a covalent bond with the drug, which may or may not be reversible [29,225]. Some examples are the EGFR/HER2 inhibitors afatinib and neratinib [226,227], the BTK inhibitor ibrutinib [228], and osimertinib, a specific inhibitor of EGFR mutated in T790M [229]. The ATP-competitive inhibitor FR148083 (5Z-7-oxozeaenol) prevents the activity of ERK1/2 with K_i_ values of 0.31 and 0.14 µM, respectively [208,230]. The crystal structure of the FR180204-ERK2 complex revealed that the residues comprising the ATP-binding pocket on ERK are essential for the interaction with this drug. 

Allosteric ERK1/2 inhibitors impede the interaction of the kinase with the protein’s interactors upstream and downstream of the signaling cascade, in a highly specific manner [29,231]. Bioinformatic approaches to identify protein domains for specific interactions represent a potential avenue for designing highly selective drugs. In the case of ERK1/2 two major domains of interaction have been recognized: The F- and the D- recruitment sites (discussed above). Early investigations in silico screened for small molecules that would recognize the polar cleft between the acidic docking domain and the Glu-Asp residues located at the F- and D- recruitment sites, both of which are exposed in the non-phosphorylated kinase [213,232,233,234]. Considering the variety of targets recognized by these two recruitment sites, the potential effects of drugs directed to these domains include a wide range of possible cellular responses. In this regard, studies have produced several candidate compounds that successfully inhibited ERK1/2-dependent phosphorylation of the target proteins p90RSK and ELK1 and inhibited proliferation of cancer cell lines in vitro [213,232,233,234]. From this screen, one of the most remarkable molecules was SF-3-030. This small molecule inhibited ERK-dependent phosphorylation of ELK1, activation of early ERK1/2-dependent genes, and impaired the proliferation in vitro of B-RAF-mutated melanoma cell lines [233]. However, due to the chemical nature of these compounds, it is necessary to further evaluate whether the responses observed are not due to off target effects [29].

Mutations on ERK2 helped to determine potential residues which may be relevant for the establishment of the kinase’s interactions and may be interesting targets for allosteric inhibitors [235]. Among these domains is the MAPK insert site, specific for the interaction with MEK1/2, which allows ERK’s phosphorylation and dimerization [236]. Mechanistic investigation is needed to understand whether impairing this dimerization domain by allosteric inhibitors affects the development of cancer [237]. However, the small molecule DEL-22379, which impairs ERK1/2 dimerization but not the kinase’s phosphorylation or catalytic activity, has an inhibitory effect on the proliferation of cultured cancer cells and xenografts [238]. Moreover, in vitro assays of MEK- and B-RAF-resistant cultured cancerous cells were sensitive to this drug [238]. Thus, there is the potential to make DEL-22379 a potential novel drug for a variety of cancers that can develop drug resistance.

Cumulative research described here speaks to the wide range of potential mechanisms to impair ERK activity, the ultimate and fundamental effector of the RAS/RAF/MEK/ERK cascade. Due to its unique structural properties, mechanisms of action, feedback regulation, and numerous substrates, ERK1/2 is an ideal candidate for developing single or combined strategies to prevent tumor progression. However, the same appealing properties of ERK as a target make the development of drugs against this kinase challenging, as the potential for off-target effects from these novel compounds is also elevated. Thus, future strategies for development of drugs against ERK should utilize comprehensive, interdisciplinary methods to prevent deleterious reactions. However, taking into account the promising outcomes of ERK inhibitors in preclinical research, ERK1/2 is an optimal target to overcome acquired drug resistance in the RAS-RAF-MEK-ERK pathway.

## 8. Conclusions

Kinases from the ERK family are able to promote EMT and enhance the cellular migration and invasion properties of tumor cells, as well as cell-cycle progression and resistance to apoptosis and drugs. ERK proteins are overexpressed and hyperactive in diverse cancer forms. Upon activation of the ERK pathway, several proteins that regulate EMT are activated, promoting tumor growth and metastasis. With ERK being the ultimate effector of the MAPK cascade, it is an ideal target to design novel therapies for patients that have developed drug resistance for other members of this signaling pathway. 

## Figures and Tables

**Figure 1 ijms-20-02885-f001:**
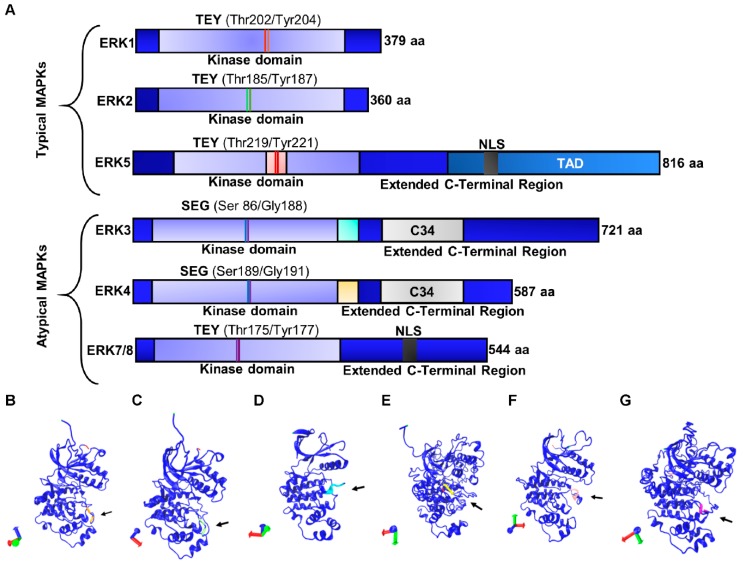
Schematic representation of ERK family members. (**A**) Domains of the typical and atypical kinases. N-terminus, C-terminus, and kinase domains are indicated. (**B**–**G**) Crystal structures of ERK proteins, depicting the phosphorylation sites (black arrows). (**B**) ERK1; the residues Thr202 and Tyr294 are shown in orange (PDB: 2ZOQ). (**C**) ERK2, Thr185, and Tyr187 are in green (PDB: 1TVO). (**D**) ERK3, Ser189, Glu190, and Gly191 are in cyan (PDB: 2I6L). (**D**) Homology model of ERK4 using I-TASSER; Ser186, Glu187, and Gly188 are in yellow. (**E**) ERK5, Thr219, Glu229, and Tyr221 are in pink (PDB: 4IC7). (**F**) Homology model of ERK8 generated with I-TASSER; Thr175 and Tyr177 are in purple.

**Figure 2 ijms-20-02885-f002:**
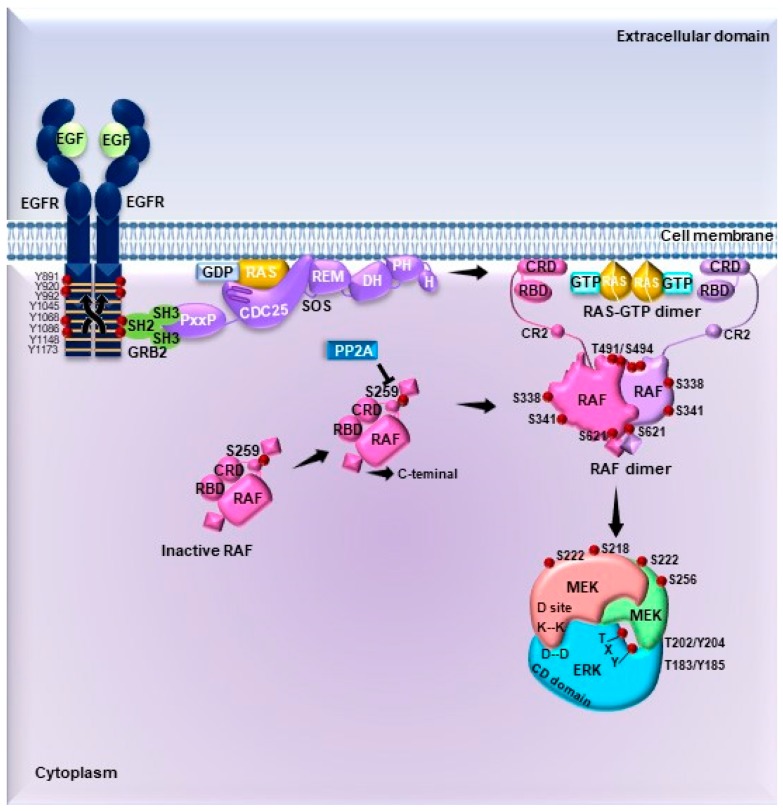
Epidermal Growth Factor Receptor-dependent activation of the MAPK signaling pathway.

**Figure 3 ijms-20-02885-f003:**
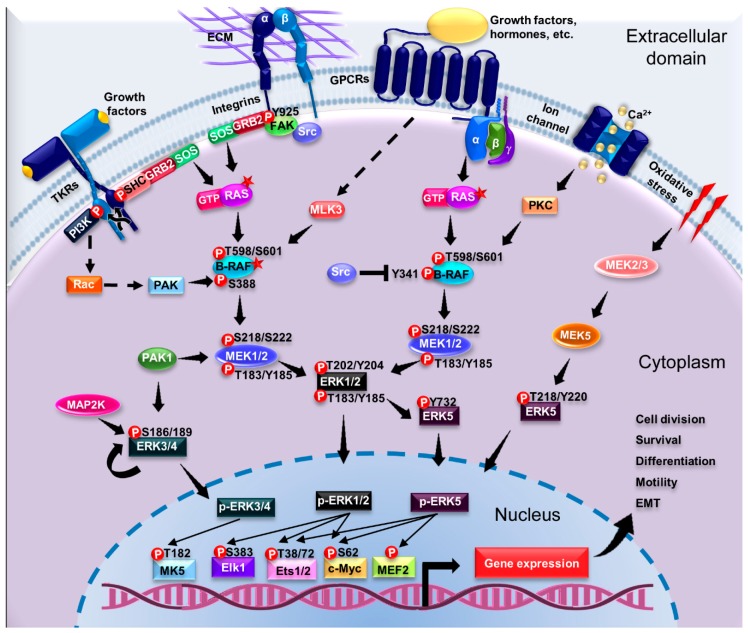
Signaling pathways driving ERK activation, effectors, and cellular processes regulated by this kinase.

**Figure 4 ijms-20-02885-f004:**
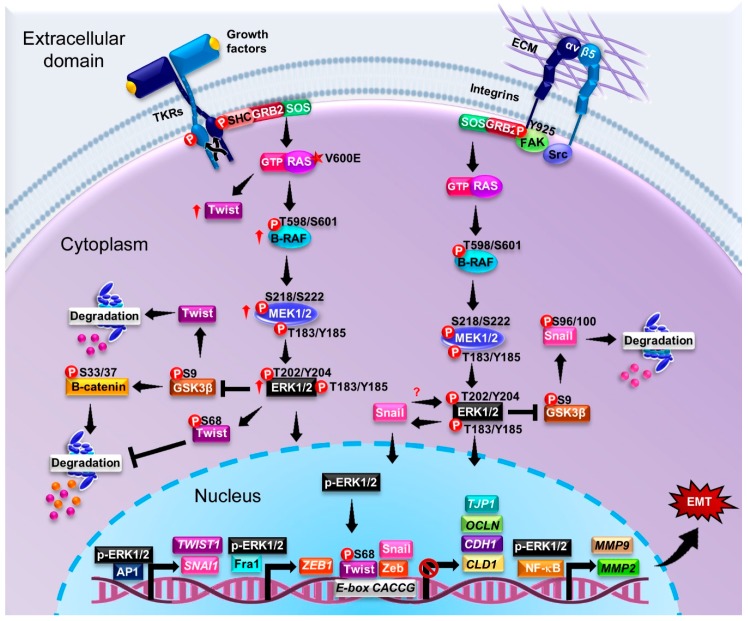
Relationship between ERK signaling and epithelial–mesenchymal transition (EMT)-associated transcription factors.

**Table 1 ijms-20-02885-t001:** ERK family members.

Protein	MAPK Family and Gene Name	UniProtKB Accession ID	Isoform	Length (Amino Acids)	Mass (Da)	Phosphorylation Sites
ERK1	MAPK3	P27361	Isoform 1 Isoform 2 Isoform 3	379 335 357	43,136 38,275 40,088	Thr202/Tyr204
ERK2	MAPK1	P28482	Isoform 1 Isoform 2	360 316	41,390 36,432	Thr185/Tyr187
ERK3	MAPK6	Q16659	–	721	82,681	Ser186/Gly188
ERK4	MAPK4	P31152	–	587	65,922	Ser189/Gly191
ERK5	MAPK7	Q13164	Isoform 1 Isoform 2 Isoform 3 Isoform 4	816 677 533 451	88,386 73,218 59,328 50,152	Thr219/Tyr221
ERK7/8	MAPK15	Q8TD08	Isoform 1 Isoform 2 Isoform 3	544 254 277	59,832 28,696 31,112	Thr175 and Tyr177

**Table 2 ijms-20-02885-t002:** ERK inhibitors.

Inhibitor	Target	Mechanism	Reference
FR180204	ERK1/2	Interacts with the ATP-binding pocket and competes with the nucleotide avoiding the kinase activation	[207]
SCH772984	ERK1/2, ERK4	ATP competitive inhibitor of ERK1/2 preventing the activation of ERK by MEK	[208,209]
MK-8353	ERK1/2	Unknown mechanism	[209,210]
VTX-11e	ERK2	Impair the phosphorylation of the kinase’s substrate p90RSK	[207]
BVD-523	ERK1/2	Unknown mechanism	[211]
“Compound 35”	ERK1/2	Binds covalently to the ATP-binding pocket preventing the kinase activation	[212]
SF-3-030	ERK2	Inhibits ERK-dependent phosphorylation of ELK1	[213]
DEL-22379	ERK1/2	Inhibits ERK dimerization without affecting phosphorylation or kinase activity	[207]

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
