# Peer review of "Extracellular-Signal Regulated Kinase: A Central Molecule Driving Epithelial–Mesenchymal Transition in Cancer"

_ijms, 2019, doi:10.3390/ijms20122885_

Round 1
Reviewer 1 Report
This is an excellent review article depicting very comprehensively the role of ERK in Caner related Epithelial Mesenchymal Transition. The article is written in a comprehensive systematic manner.
Line 68: This decreased adhesiveness is due to AN HINDERED production 8 of canonical epithelial markers such as E-cadherin, cytokeratins, and ZO-1, as well as the production 69 of ECM-degrading enzymes, and ultimately the reorganization of the cytoskeleton.
Comment: This sentence needs reconstruction to make the meaning clear.
Line 75: In addition to its participation in normal physiological processes, the EMT can be triggered by many oncogenic signaling pathways, hypoxia, and signals of tumor microenvironment.
Comment: The phenomenon of EMT is also triggered by hyproxia and reactive oxygen species mediated injury. PMID: 24375795.
Line 372: ERK5 is activated in response to serum, growth factors.
Comment: No comma required after serum.
Section 7. ERK as therapeutic target in cancer:
Comment: A separate Table of therapeutic targets with mechanism should be very useful
Author Response
Response to Reviewer 1 Comments
Point 1. This is an excellent review article depicting very comprehensively the role of ERK in Caner related Epithelial Mesenchymal Transition. The article is written in a comprehensive systematic manner.
Response 1: We thank the reviewer for his positive evaluation of our manuscript, and for his/her comments and suggestions indicated below.
Point 2. Line 68: This decreased adhesiveness is due to AN HINDERED production 8 of canonical epithelial markers such as E-cadherin, cytokeratins, and ZO-1, as well as the production 69 of ECM-degrading enzymes, and ultimately the reorganization of the cytoskeleton.
Comment: This sentence needs reconstruction to make the meaning clear.
Response 2: The indicated sentence has been fixed as requested.
Point 3. Line 75: In addition to its participation in normal physiological processes, the EMT can be triggered by many oncogenic signaling pathways, hypoxia, and signals of tumor microenvironment.
Comment: The phenomenon of EMT is also triggered by hyproxia and reactive oxygen species mediated injury. PMID: 24375795.
Response 3: We have corrected the sentence and added the suggested reference
Pount 4. Line 372: ERK5 is activated in response to serum, growth factors.
Comment: No comma required after serum.
Response 4: Deleted
Point 5. Section 7. ERK as therapeutic target in cancer:
Comment: A separate Table of therapeutic targets with mechanism should be very useful.
Response 5: We appreciate this valuable suggestion. A new Table 2 has been included to summarize this point. Thank you

Reviewer 2 Report
In this review whose title is “Extracellular-signal Regulated Kinase: a Central Molecule Driving Epithelial-Mesenchymal Transition in Cancer” authors highlight the many aspect of
ERK molecule and kinase namely structure, isoforms, atypical isoforms, substrates and regulation of activity and, besides, the involvement of ERKs in Epithelial-Mesenchymal Transition.
Finally, in a specific section authors highlighted the experimental evidence obtained from different cancer cell lines, animal models and biopsies of patients, which have established the relationship between ERK and EMT in different types of cancer. This is very useful and interesting to be updated on this special field.
To my view this review has an overall positive impact in cancer signal transduction and is interesting for many readers.
Author Response
Response 1. Comments and Suggestions for Authors
In this review whose title is “Extracellular-signal Regulated Kinase: a Central Molecule Driving Epithelial-Mesenchymal Transition in Cancer” authors highlight the many aspect of ERK molecule and kinase namely structure, isoforms, atypical isoforms, substrates and regulation of activity and, besides, the involvement of ERKs in Epithelial-Mesenchymal Transition.
Finally, in a specific section authors highlighted the experimental evidence obtained from different cancer cell lines, animal models and biopsies of patients, which have established the relationship between ERK and EMT in different types of cancer. This is very useful and interesting to be updated on this special field.
To my view this review has an overall positive impact in cancer signal transduction and is interesting for many readers.
Response 1: We thank the reviewer for his/her valuable opinion on our work and are grateful for the positive comments made by the reviewer.
